# Low Dosed Curcumin Combined with Visible Light Exposure Inhibits Renal Cell Carcinoma Metastatic Behavior in Vitros

**DOI:** 10.3390/cancers12020302

**Published:** 2020-01-28

**Authors:** Jochen Rutz, Sebastian Maxeiner, Saira Justin, Beatrice Bachmeier, August Bernd, Stefan Kippenberger, Nadja Zöller, Felix K.-H. Chun, Roman A. Blaheta

**Affiliations:** 1Department of Urology, Goethe-University, 60590 Frankfurt am Main, Germany; Sebastian.Maxeiner@kgu.de (S.M.); Justinsaira@hotmail.com (S.J.); Felix.Chun@kgu.de (F.K.-H.C.); Blaheta@em.uni-frankfurt.de (R.A.B.); 2Institute of Laboratory Medicine, University Hospital, Ludwig-Maximilians-University, 80539 Munich, Germany; Beatrice.bachmeier@tum.de; 3Department of Dermatology, Venereology, and Allergology, Goethe-University, 60590 Frankfurt am Main, Germany; Bernd@em.uni-frankfurt.de (A.B.); Stefan.Kippenberger@kgu.de (S.K.); Nadja.Zoeller@kgu.de (N.Z.)

**Keywords:** curcumin, renal cell cancer, tumor adhesion, tumor migration, integrins

## Abstract

Recent documentation shows that a curcumin-induced growth arrest of renal cell carcinoma (RCC) cells can be amplified by visible light. This study was designed to investigate whether this strategy may also contribute to blocking metastatic progression of RCC. Low dosed curcumin (0.2 µg/mL; 0.54 µM) was applied to A498, Caki1, or KTCTL-26 cells for 1 h, followed by exposure to visible light for 5 min (400–550 nm, 5500 lx). Adhesion to human vascular endothelial cells or immobilized collagen was then evaluated. The influence of curcumin on chemotaxis and migration was also investigated, as well as curcumin induced alterations of α and β integrin expression. Curcumin without light exposure or light exposure without curcumin induced no alterations, whereas curcumin plus light significantly inhibited RCC adhesion, migration, and chemotaxis. This was associated with a distinct reduction of α3, α5, β1, and β3 integrins in all cell lines. Separate blocking of each of these integrin subtypes led to significant modification of tumor cell adhesion and chemotactic behavior. Combining low dosed curcumin with light considerably suppressed RCC binding activity and chemotactic movement and was associated with lowered integrin α and β subtypes. Therefore, curcumin combined with visible light holds promise for inhibiting metastatic processes in RCC.

## 1. Introduction

An estimated 18.1 million new patients worldwide were diagnosed with cancer in 2018, and of these, 9.6 million people died [1]. Characteristics of cancer are the loss of normal cell communication, unlimited cell growth, increased mobility, and the suppression of apoptosis [2]. Migration and motile spread are critical steps in tumor dissemination and progress. As with most tumors, metastasis plays an important role in renal cell carcinoma (RCC) and is the main cause of mortality [3]. Metastases infiltrate bones, lungs, lymph nodes, and less often brain and liver. At first diagnosis, one third of patients already suffer from lymph node and/or organ metastases [4] with another 20 to 30% developing metastases during therapy [5]. Without treatment, the probability of survival one year after diagnosis of metastasis is only about 50% [6], and the chances of recovery are poor. More than 5000 people died in Germany due to renal cell carcinoma in 2011 [7] and in a disseminated stage only palliative therapy can be provided. The incidence has been increasing in recent decades and has reached a constant level since the 1990s. The mortality is about 8/100,000 for men and 3/100,000 for women [8]. Targeted therapies have been introduced to improve the survival rate, but the prognosis for survival has hardly changed. Aspiring to active involvement, dissatisfaction with conventional medicine, and the hope to reduce unwanted side effects has made patients turn to complementary and alternative medicine (CAM). These approaches range from yoga to mind stimulating music to application of phytopharmacological agents. Up to 50% of cancer patients in Europe use CAM in addition to, or in place of, conventional medicine [9,10].

The natural compound curcumin (Figure 1) is a component of turmeric, a yellow-orange pigment harvested from the rhizomes of the plant Curcuma longa. Aside from its use as a spice in curry powder, anti-inflammatory, anti-oxidative, and anti-tumorigenic qualities have been demonstrated in vitro and in vivo [11,12,13], making it interesting for clinical application. Diverse biochemical processes and pathways associated with carcinogenesis are affected and modulated by curcumin [14]. In prostate, lung, breast, and colorectal cancers it has been shown [15] that curcumin affects growth and proliferation by inhibiting cell cycle progression, angiogenesis, and the expression of anti-apoptotic proteins [14,16,17].

However, due to poor water solubility, low absorption, rapid metabolism and elimination, curcumin has low bioavailability, hampering its clinical use [19,20]. To improve bioavailability several approaches have been employed, such as wrapping lipophilic curcumin in liposomes, micelles, solid lipid nanoparticles, or polymer conjugates [21,22,23]. Likewise a range of analogues play a role in enhancing the bioavailability of curcumin [24]. Although some approaches have been successful, further improvement in bioavailability would be beneficial [22,25].

The present study continues a previous investigation on an RCC cell model demonstrating that exposing curcumin treated cells to visible light considerably enhances curcumin’s potential to suppress tumor growth and proliferation [26]. Since metastasis, rather than that the growth of the primary tumor is the main cause of mortality, a therapeutic strategy blocking metastatic progression was investigated here. For this purpose, the influence of low dosed curcumin combined with visible light on adhesion and chemotaxis, as well as on intra- and extracellular integrin expression and signaling, was evaluated using a panel of three RCC cell lines.

## 2. Results

### 2.1. Curcumin Uptake

Curcumin uptake studies were carried out using 4 µg/mL curcumin instead of 0.2 µg/mL. This was necessary, since the fluorescence intensity of 0.2 µg/mL curcumin was too low to provide high quality images for confocal microscopy and optimum fluorescence detection by FACS analysis. Using 4 µg/mL, curcumin was rapidly incorporated into the cells following administration. In A498 cells, maximum fluorescence intensity was noted after 50 min, whereas a plateau phase was reached after 40 min in Caki1 and already after 20 min in KTCTL-26 cells (Figure 2A). Confocal microscopy demonstrated homogenous cytoplasmic distribution of curcumin in all three cell lines with accumulation along the nuclear membrane (Figure 2B, solid arrows). Curcumin was also visualized within the nucleus (Figure 2B, dashed arrows).

### 2.2. Tumor Cell Adhesion and Binding Behavior

Adhesion of all three cell lines to HUVECs was blocked by combining 0.2 µg/mL curcumin with visible light (Figure 3A). The number of cells attached after 2 h (mean adhesion/mm^2^, controls versus curcumin^Light^) was: 62.8 ± 5.3 versus 20.6 ± 4.1 for A498; 40.6 ± 7.0 versus 9.2 ± 2.0 for Caki1; 40.6 ± 6.7 versus 14.4 ± 3.4 for KTCTL-26. Light exposure alone or curcumin alone had no more effect on adhesion than the addition of cell medium as a control. A similar response was seen in the binding behavior to immobilized collagen. Neither curcumin nor light exposure alone led to significant alterations of A498, Caki1, or KTCTL-26 cell binding, compared to the untreated controls. Combined use of curcumin and light was associated with a distinct attenuation in the tumor cell attachment rate, with maximum effects exerted on Caki1 cells (17.8 ± 6.0%, compared to the 100% control; Figure 3B).

### 2.3. Chemotaxis and Migration

Figure 4A shows that neither treatment with light nor cultivation of the tumor cells with curcumin influenced chemotactic movement towards a serum gradient. A distinct down-regulation of chemotaxis was induced when the tumor cells were exposed to low-dosed curcumin with light. This response became evident in all three cell culture systems with the order KTCTL-26 (12.7 ± 3.6%) > Caki1 (20.9 ± 5.0%) > A498 (47.6 ± 9.6%), each compared to the 100% control; Figure 4A.

Tumor cell migration through a collagen matrix towards a serum gradient was also evaluated. Curcumin or light alone did not alter the trans-migration rate, whereas curcumin^Light^ did (Figure 4B). Migration was nearly completely abrogated in KTCTL-26 cells (3.4 ± 4.2%, compared to the 100% control).

### 2.4. FACS Analysis of Iintegrin Surface Expression

Different integrin expression patterns were apparent for the different tumor cell lines. In A498 cells, the integrin subtypes α3, β1, and β3 were expressed to the highest extent at the cell surface. Distinct fluorescence intensity was also recorded for α1 and α5. The subtypes α2, α4, and α6 were detected moderately, whereas β4 was not expressed at all (Figure 5).

Caki1 cells were characterized by a very strong expression of α3 and β1, and a strong expression of α5 and β3. α2 and α6, as well as β4, were also present at the surface membrane. α1 and α4 were only marginally detectable (Figure 5).

Similar to Caki1, α3 and β1 were also expressed to the highest extent on KTCTL-26 cells. α5 and β3 were distinctly detectable. α1, α2, α6, and β4 were also detectable (Figure 5). α4 was barely detectable.

The integrin expression level was not modulated by curcumin or light, when applied separately, but significant alterations were evoked by curcumin^Light^. The surface expression of α3, α5, β1, and β3 was diminished in all three tumor cell lines (A498: α3: −14.0 ± 1.3%, α5: −57.1 ± 3.7%, β1: −23.8 ± 1.5%, β3: −56.7 ± 4.4%; Caki1: α3: −15.9 ± 0.8%, α5: −18.0 ± 1.2%, β1: −37.3 ± 2.7%, β3: −20.4 ± 1.5%; KTCTL-26: α3: −41.1 ± 3.7%, α5: −40.2 ± 2.0%, β1: −45.8 ± 4.0%, β3: −25.6 ± 3.2%; each compared to the 100% control), while α1 was suppressed in A498 (−48.7 ± 2.4%) and KTCTL-26 cells (−37.0 ± 2.2%). α2 (−22.1 ± 1.8%) and α6 (−29.1 ± 1.8%) were exclusively down-regulated on KTCTL-26 (Figure 6). The influence of curcumin and/or light on α4 and β4 was not evaluated, since α4 was not expressed on all cell lines and β4 was not expressed on A498 and only slightly expressed on Caki1 and KTCTL-26 cells.

### 2.5. Western Blot Analysis

Figure 7 illustrates changes in the integrin protein content including ILK, FAK, and pFAK after exposure to curcumin, light, or curcumin^Light^ (whole blots are shown in Appendix A). Figure 8 shows the respective pixel density data (pixel density and *p*-values are shown in Appendix A). Exposing the tumor cells to visible light did not lead to distinct protein modifications, excepting α2 and β1, both of which were up-regulated in KTCTL-26 cells, and pFAK which was diminished in KTCTL-26 and elevated in A498 cells, compared to the untreated controls. Curcumin alone also had no specific effect on protein expression, excepting β3, which was up-regulated in Caki1 cells, and α5, which was enhanced in KTCTL-26 cells. In contrast, a strong response became evident when the tumor cell lines were treated with curcumin^Light^. Here, compared to the controls, exposure to light alone or treatment with curcumin alone, the following proteins were reduced: A498-α1, -β1, -β3; Caki1-α5, -β1; KTCTL-26-α1, -α2, -β1, -β3. ILK, FAK, and pFAK were suppressed in all cell lines when exposed to curcumin^Light^.

### 2.6. Integrin Blockage

The FACS analysis demonstrated that curcumin^Light^ induces the loss of α3, α5, β1, and β3 in all tumor cell lines. Therefore, the surface expression of α3, α5, β1, and β3 was blocked to evaluate the physiological and pathological relevance of these integrin subtypes on adhesion (Figure 9A) and chemotaxis (Figure 9B).

Blocking integrin α3 was associated with reduced A498 and Caki1 adhesion (A498: −20.9 ± 1.9%; Caki1: −21.2 ± 2.0%) and diminished A498 and KTCTL-26 chemotaxis (A498: −17.5 ± 2.3%; KTCTL-26: −78.3 ± 8.6%). Blocking α5 down-regulated A498 and KTCTL-26 adhesion (A498: −16.0 ± 1.3%; KTCTL-26: −57.6 ± 7.3%) and chemotaxis (A498: −26.4 ± 2.4%; KTCTL-26: −27.0 ± 5.0%), whereas chemotaxis of Caki1 was enhanced (+33.7 ± 4.1%). Surface blocking of β1 suppressed adhesion of all cell lines in the order Caki1 (−89.7 ± 6.7%) > A498 (−57.1 ± 1.0%) > KTCTL-26 (−21.3 ± 3.6%), and diminished Caki1 chemotaxis (−19.7 ± 3.3%). β3 blockade down-regulated KTCTL-26 adhesion (−12.3 ± 1.0%) and A498 chemotaxis (−23.5 ± 1.9%).

## 3. Discussion

Exposing RCC cells to visible light significantly enhanced curcumin’s potential to block adhesion and migration. While low dosed curcumin alone induced no alterations in tumor-endothelial or tumor-matrix interaction, the combination of curcumin plus light did. Photodynamic properties of curcumin are well documented, though the mechanistic background is not fully understood. In general, irradiation of a photodynamic molecule with a particular wavelength shifts electrons to higher energy orbitals. This singlet state is unstable, and the electrons return to their ground state by emitting light or heat. However, changes in electron spin can also shift a photodynamic molecule to the triplet state, which then triggers two reactions. The Type 1 reaction produces free radicals and, due to an interaction with oxygen, reactive oxygen species (ROS). The Type 2 reaction results in singlet oxygen that can interact with specific intracellular molecules [27]. Whether this mechanism also holds true for curcumin is not yet clear. Laubach et al. assumed a shift in the cellular redox balance by boosting H_2_O_2_ generation [28]. Bruzell et al. speculated that curcumin may photo-generate reduced forms of molecular oxygen [29]. In our own pilot experiments, treating curcumin with light prior to application did not enhance the anti-tumor effect of curcumin, compared to the application of curcumin without light. In line with this, intra-peritoneal injections of curcumin with or without light induced the same effect on nerve injury repair in a mouse model [30]. The phenomenon could be attributed to the unstable excitation state of irradiated curcumin with a very short half-life, although other mechanisms cannot be excluded. Bernd has suggested that a light-dependent energy transfer via curcumin may enhance the influence of this compound on tumor relevant protein functions [18]. Niu et al. assumed photo-activation to be an essential amplification factor when taking advantage of curcumin at low concentrations [31].

Low-dosed curcumin (0.2 µg/mL) combined with light profoundly blocked RCC cell adhesion to HUVEC, while even 5 μM of free curcumin without light could not alter the binding of prostate cancer cells to HUVEC [32]. In the present study, since the number of attached RCC cells was maximally reduced after 30 min, with no further diminishment at 1 or 2 h, curcumin^Light^ seems to exert its effect in the initial attachment phase. A strong benefit of adding light to a low curcumin concentration (0.2 µg/mL) was also evident in the tumor cell-matrix attachment, where curcumin^Light^ (but not curcumin alone) down-regulated the binding of all cell lines to immobilized collagen. Herman et al. have demonstrated adhesion blocking effects of curcumin on prostate cancer cells [33] and others have demonstrated that curcumin suppresses binding of esophageal [34], skin [35], or breast cancer [36] to extracellular matrix proteins. In all cases, high concentrations of 5−50 µM curcumin were necessary to exert therapeutic efficacy. Until now, the relevance of curcumin to RCC adhesive processes had not been documented.

The benefit of visual light on curcumin’s bioavailability was also seen in regard to chemotaxis and migration. 0.2 µg/mL curcumin^Light^, but not 0.2 µg/mL curcumin alone, profoundly reduced motile crawling of all three RCC cell lines. The invasion blocking effect of curcumin is important, since once metastasized, cancer is difficult to treat, and the extent of metastasis rather than the primary cancer determines survival. Therefore, application of curcumin^Light^ might be an innovative concept to accompany established RCC treatment protocols. The relevance of curcumin alone to act on tumor cell invasion has already been shown on other tumor entities, whereby curcumin concentrations of 10 µM, 15 µM, 50 µM, or higher have been applied to stop invasion of gastric [37], breast [38], prostate [39], or hepatic cancer cells [40]. Ongoing experiments should, therefore, deal with the question of whether the beneficial effects of light exposure on curcumin may also hold true for these tumor types.

When interpreting the influence of curcumin^Light^ on adhesion and chemotaxis, it is notable that curcumin^Light^ considerably blocked adhesion of Caki1, whereas KTCTL-26 adhesion was only slightly suppressed. In contrast, migration properties of KTCTL-26 were suppressed to a maximum, migration of Caki1 was suppressed moderately. Due to these differences, lowered migration does not seem to exclusively be just a consequence of a reduced attachment rate. Rather, curcumin^Light^ is involved in both the regulation of the mechanical tumor cell-matrix contact and modulation of cytoskeletal structures. Indeed, curcumin has been shown in tumor cell models to disorganize the architecture of actin microfilaments, leading to destabilization and a decrease in F-actin polymerization [41,42]. Dhar et al. assumed an allosteric effect in which curcumin binding at the "barbed end" of actin is transmitted to the "pointed end," where conformational changes disrupt interactions with the adjacent actin monomer to interrupt filament formation [43]. There is also evidence that curcumin stops the physical interaction of cortactin with p120 catenin, which then may inhibit migration [44].

Beyond intracellular components, membrane proteins expressed on the cell surface are also relevant for controlling cell movement. Alterations of the integrin α- and β-expression pattern have been closely associated with altered metastatic activity [45]. Therefore, integrins are considered to be highly relevant treatment targets [46]. The data presented here point to distinct changes of particular integrin subtypes in the presence of curcumin^Light^, but not in the presence of curcumin alone. Of all integrin members evaluated, surface expression of four subtypes were modified in the same manner in all cell lines; α3, α5, β1, β3 were all down-regulated by curcumin^Light^. Since the intracellular α3 protein content was neither reduced in Caki1, nor in KTCTL-26 and A498 cells, α3 might be shed from the surface without intracellular alteration. Shedding may also be relevant for α5 (KTCTL-26) and β1 (KTCTL-26). It is hypothesized that the β3 protein increase in Caki1 is caused by curcumin^Light^ inducing a translocation from the cell surface to the cytoplasm.

The relevance of integrins in tumor progression is not completely understood. Integrin α3 is thought to be closely associated with the capacity of RCC for local and distant spread [47]. The same attribute has been linked to integrin β3 [48], and based on clinical specimens from tumor patients, α3 as well as β3 have been proposed as potential prognostic markers [49,50]. Evidence indicates that the integrin subtype α5 correlates with poor survival [51]. In fact, α5 is the most highly expressed integrin in RCC tissue, compared with adjacent normal renal tissue, and knocking down α5 has been shown to significantly reduce cell migration [52].

Integrin β1 also plays an important role in the development of RCC tumors and advanced RCC with metastasis [53], the observation of which has led to the development of volociximab, an anti-α5β1 integrin monoclonal antibody [54]. The suppressive effect of low dosed curcumin plus light on the integrins α3, α5, β1, and β3 on RCC adhesion, chemotaxis, and migration could (at least in part) be attributed to inhibition of these integrins, which when blocked were shown to inhibit RCC binding and spreading. The inhibition of adhesion or chemotaxis depended on the cell line. β1, which was strongly reduced on Caki1 cells by curcumin^Light^, was also prominently involved in regulating adhesion in this cell line. The integrin α5 subtype, the major regulator of KTCTL-26 adhesion, was also considerably suppressed by curcumin^Light^ in KTCTL-26 cells. α3 integrin served as a dominant element in reducing KTCTL-26 chemotaxis. The same integrin was also considerably suppressed by curcumin^Light^. These data could indicate that metastatic tumor progression is controlled by different integrin members, depending on the tumor differentiation status, and that these specific integrins act as main targets for curcumin^Light^. This is, however, speculative. A498 chemotaxis depended equally well on α3, α5, and β3. It must also be considered that quantitative alteration of the integrin surface expression, but not activity, was evaluated. Whether integrin loss is associated with a similar loss of activity cannot be judged.

Suppressed integrin α3 was associated with inhibited adhesion in A498 and Caki1 cells but not in KTCTL-26 cells. Chemotaxis was inhibited in A498 and KTCTL-26, but not in Caki1 cells. Suppression of β3 exclusively prevented A498 adhesion and KTCTL-26 chemotaxis. Blockade of α5 coupled to an increased chemotaxis rate of Caki1 is paradoxic and difficult to explain, since curcumin^Light^ evoked α5 inhibition would be expected to contribute to increased motile behavior, which was not the case. Rather, chemotaxis of Caki1 was considerably blocked by curcumin^Light^. The extent to which α5 was diminished in Caki1 by curcumin^Light^ was only −20%, compared to an α5 reduction in A498 (−60%) and KTCTL-26 cells (−40%). Speculatively, the moderate alteration of Caki1’s α5 surface level by curcumin^Light^ is of minor relevance for adhesion and migration. Counter regulation should be considered, and this α5 behavior in Caki1 may point to resistance induction.

Aside from the paradoxical role of α5 in Caki1 cells, curcumin^Light^ is shown here to act on a set of integrin receptors which, in combination, profoundly blocks metastatic progression in vitro. This indicates that the complex process of metastasis is not controlled by only one particular integrin subtype. Rather, several integrins seem to be regulatory elements driving the invasion cascade forward. Consequently, blocking a set of relevant integrin members, as curcumin^Light^ did, might be more effective than blocking just a single integrin.

The therapeutic potential of curcumin is also reflected by its deactivation of FAK. FAK serves as a prominent linker molecule, connecting integrin related signaling with pro-mitogenic and pro-migratory pathways including the Ras-ERK and PI3K/AKT pathway [55]. Performing a mass spectrometry-based system-wide survey of tyrosine phosphorylation in clear cell and papillary RCC human tumors, distinct FAK phosphorylation has been found in all tumors [56]. FAK may also mediate resistance towards the tyrosine kinase inhibitor sorafenib in RCC patients [57]. This opens the possibility that light exposure to curcumin treated RCC cells might not only be an innovative strategy to fight metastatic progression but also to enhance or prolong the response towards a tyrosine kinase inhibitor-based regimen. Curcumin combined with sorafenib or sunitinib has already been demonstrated to synergistically inhibit cancer growth and metastasis in vitro and in vivo [58,59].

The technical aspect of curcumin–light application has been addressed. Introducing an optical fiber into RCC tumors in mice with subsequent laser illumination of the vascular-acting photosensitizer WST11 at 750 nm or multispectrally at 700–800 nm has been shown to induce significant necrosis in RCC tissue [60]. Kroeze et al. have suggested using the photosensitizer mTHPC (meso-tetra(hydroxyphenyl)chlorin), which targets both vasculature and tissue and, therefore, may produce a strong combined effect [61]. Exposing the tumor bed to light after tumor resection and curcumin administration has also been discussed in regard to eliminating invisible micro-metastases [62].

## 4. Materials and Methods

### 4.1. Cell Culture

Renal carcinoma Caki1 and KTCTL-26 cell lines, both derived from a clear cell renal cell carcinoma and von Hippel-Lindau (VHL) positive, were purchased from LGC Promochem (Wesel, Germany). A498 cells with disrupted VHL function were derived from Cell Lines Service (Heidelberg, Germany). The tumor cells were grown and subcultured in RPMI 1640 medium supplemented with 10% fetal calf serum (FCS), 1% Glutamax (all Gibco/Invitrogen, Karlsruhe, Germany), 2% HEPES (2-(4-(2-Hydroxyethyl)-1-piperazinyl)-ethansulfonsäure) buffer and 1% penicillin/streptomycin (both Sigma-Aldrich, München, Germany), at 37 °C in a humidified 5% CO_2_ incubator. Subcultures from passages 5–30 were selected for experimental use.

Human umbilical vein endothelial cells (HUVEC), isolated from human umbilical veins, were grown in Medium 199 (M199; Biozol, Munich, Germany), 10% FCS, 10% pooled human serum, 20 μg/mL endothelial cell growth factor (Boehringer, Mannheim, Germany), 0.1% heparin, 100 ng/mL gentamycin and 20 mM HEPES-buffer. Subcultures from passages 2 to 6 were selected for experimental use.

### 4.2. Drug Dosage and Light Exposure

Curcumin (Biomol, Hamburg, Germany) was stored at −20 °C and diluted prior to use in cell culture medium to a final concentration of 0.2 µg/mL (0.54 µM). 4 µg/mL curcumin was used to provide high quality images for confocal microscopy and optimum fluorescence detection by FACS analysis. Cells were treated with curcumin for 1 h and then exposed to visible light for 5 min with 5500 lx (curcumin^Light^; 10 × 40 W lamps, distance 45 cm, emission spectrum: 400–550 nm) using a Waldmann UV 801AL system (Waldmann, Villingen-Schwenningen, Germany) [18]. To prevent bias effects by the phenol red containing RPMI 1640 based cell culture medium, tumor cells were transferred to phenol red free PBS (phosphate-buffered saline) (Sigma-Aldrich) during light exposure. Thereafter, PBS was replaced by RPMI 1640 and supplements. Control cell cultures received PBS for 5 min without light exposure. To evaluate the effects of low dosed curcumin and light alone, two respective additional controls were employed; tumor cells exposed to light but not to curcumin, and tumor cells exposed to curcumin but no light. Following light exposure (including all controls), tumor cells were allowed to recover in complete cell culture medium for 24 h before starting adhesion and migration experiments.

### 4.3. Cellular Curcumin Uptake

5 × 10^4^ RCC cells were plated on 6-well multiplates (Sarstedt, Nümbrecht, Germany) and, when grown to sub-confluency, incubated with 4 µg/mL curcumin for different time periods ranging from 10 to 60 min at 37 °C. Thereafter, the tumor cells were detached, washed three times with PBS (Ca^2+^ and Mg^2+^) and subsequently added to FACS-buffer (PBS + 0.5% bovine serum albumin, BSA) at 0.5 × 10^5^ cells/mL. Fluorescence intensity (mean fluorescence units, MFU) of curcumin exposed versus non-exposed cells was then measured by a FACS Canto (BD Biosciences, Heidelberg, Germany) at an absorption of 485 nm and emission of 514 nm.

### 4.4. Intracellular Distribution of Curcumin

To evaluate intracellular localization of curcumin, tumor cells were incubated with 4 µg/mL curcumin for 60 min, washed with PBS, fixed in cold (−20 °C) methanol/acetone (50/50 v/v) and then washed with blocking buffer (0.5% BSA in PBS). To prevent photobleaching of curcumin, tumor cells were embedded in Vectashield mounting medium including DAPI (Biozol, Munich, Germany), and viewed using a confocal laser scanning microscope (Zeiss, Oberkochen, Germany, equipped with Zen imaging software) with a plan-neofluar × 63/1.3 oil immersion objective.

### 4.5. Tumor Cell Endothelial Cell Interaction

To evaluate tumor cell adhesion, HUVEC were transferred to 6-well multiplates in complete HUVEC medium. Once the cells had reached confluence, A498, Caki-1, or KTCTL-26 cells were detached from the culture flasks by accutase treatment (PAA Laboratories, Cölbe, Germany), and 0.5 × 10^6^ cells were added to the HUVEC monolayer. After 0.5, 1, or 2 h, non-adherent tumor cells were washed off using warmed (37 °C) PBS+ (Ca^2+^ and Mg^2+^). The remaining cells were fixed with 1% glutaraldehyde. Adherent tumor cells were then counted in five different observation fields of a defined size (5 × 0.25 mm^2^) using a phase contrast microscope and the mean cellular adhesion rate was calculated.

### 4.6. Attachment to Immobilized Collagen

Six-well plates were coated with collagen G (extracted from calfskin, consisting of 90% collagen type I and 10% collagen type III; Biochrom, Berlin, Germany; diluted to 400 µg/mL in PBS) overnight at 4 °C. Plastic dishes served as background control. Subsequently, plates were incubated for one h with 1% BSA in PBS to block nonspecific cell adhesion. 0.5 × 10^6^ tumor cells were then added to each well and allowed to attach for 60 min at 37 °C. Subsequently, non-adherent tumor cells were washed off, the remaining adherent cells were fixed with 1% glutaraldehyde and counted microscopically. The mean cellular adhesion rate, defined by adherent cells_coated well_ - adherent cells_background_, was calculated from five different observation fields (5 × 0.25 mm^2^).

### 4.7. Chemotaxis and Migration

Serum induced chemotactic movement was examined using 6-well Transwell chambers (Greiner, Frickenhausen, Germany) with 8 µm pores. 0.5 × 10^6^ A498, Caki1, or KTCTL-26 cells/mL were placed in the upper chamber in serum-free medium. The lower chamber contained 10% serum. To evaluate cell migration, Transwell chambers were pre-coated with collagen (400 µg/mL) and tumor cells then added. After 24 h incubation, the upper surface of the Transwell membrane was gently wiped with a cotton swab to remove cells that had not migrated. Cells that had moved to the lower surface of the membrane were stained using hematoxylin and counted microscopically. The mean chemotaxis and migration rate were then calculated from five different observation fields (5 × 0.25 mm^2^).

### 4.8. Integrin Surface Expression

Tumor cells were washed in blocking solution (PBS, 0.5% BSA) and then incubated for 60 min at 4 °C with phycoerythrin (PE)-conjugated monoclonal antibodies directed against the following integrin subtypes: anti-α1 (mouse IgG1; clone SR84; #559596), anti-α2 (mouse IgG2a; clone 12F1-H6; #555669), anti-α3 (mouse IgG1; clone C3II.1; #556025), anti-α4 (mouse IgG1; clone 9F10; #555503), anti-α5 (mouse IgG1; clone IIA1; #555617), anti-α6 (mouse IgG2a; clone GoH3; #555736), anti-β1 (mouse IgG1; clone MAR4; #555443), anti-β3 (mouse IgG1; clone VI-PL2; #555754) or anti-β4 (rat IgG2b; clone 439-9B; #555720) (all from BD Pharmingen, Heidelberg, Germany). Integrin expression of tumor cells was then measured using a FACScan (BD Biosciences; FL-2H (log) channel histogram analysis; 1 × 10^4^ cells/scan) and expressed as mean fluorescence units (MFU). Mouse IgG1-PE (MOPC-21; #555749), IgG2a-PE (G155-178; #555574), and rat IgG2b-PE (R35-38; #555848; all from BD Biosciences) were used as isotype controls.

### 4.9. Western Blotting

To investigate integrin content, tumor cell lysates were applied to a 7–12% polyacrylamide gel (depending on the protein size) and electrophoresed for about 90 min at 100 V. The protein was then transferred to nitrocellulose membranes. After blocking with non-fat dry milk for 1 h, the membranes were incubated overnight with the following antibodies: integrin α1 (rabbit, polyclonal, 1:1,000; #AB1934; Chemicon/Millipore GmbH, Schwalbach, Germany), integrin α2 (mouse IgG1, 1:250, clone 2; #611017; BD Biosciences), integrin α3 (rabbit, polyclonal, 1:1000; #AB1920; Chemicon/Millipore GmbH), integrin α4 (mouse, 1:200, clone: C-20; #sc-6589; Santa Cruz Biotechnology, Inc., Santa Cruz, CA, USA), integrin α5 (mouse IgG2a, 1:5000, clone 1; #610634; BD Biosciences), integrin α6 (rabbit, 1:200, clone H-87; #sc-10730; Santa Cruz Biotechnology, Inc.,), and integrin β1 (mouse IgG1, 1:2500, clone 18; #610468), integrin β3 (mouse IgG1, 1:2500, clone 1; #611141) and integrin β4 (mouse IgG1, 1:250, clone 7; #611233) (all from BD Biosciences). HRP-conjugated goat anti-mouse IgG and HRP-conjugated goat anti-rabbit IgG (both 1:5000; Upstate Biotechnology, Lake Placid, NY, USA) served as secondary antibodies. Additionally, integrin-related signaling was explored by anti-integrin-linked kinase (ILK) (clone 3, dilution 1:1000; #611803), anti-focal adhesion kinase (FAK) (clone 77, dilution 1:1000; #610088), and anti-p-specific FAK (pY397; clone 18, dilution 1:1000; #611807) antibodies (all from BD Biosciences). HRP-conjugated goat-anti-mouse IgG (dilution 1:5000; Upstate Biotechnology) served as the secondary antibody. The membranes were briefly incubated with ECL detection reagent (ECL™; Amersham, GE Healthcare, München, Germany) to visualize the proteins and then analyzed with the Fusion FX7 system (Peqlab, Erlangen, Germany). β-actin (1:1000; Sigma-Aldrich) served as the internal control.

Gimp 2.8 software was used to perform pixel density analysis of the protein bands. The ratio of protein intensity/β-actin intensity was calculated and expressed as percentage difference, related to controls set to 100%.

### 4.10. Blocking Experiments

To determine whether the integrins α3, α5, β1, and β3 impact metastatic spread, A498, Caki-1, or KTCTL-26 cells were incubated for 60 min with 10 µg/mL function-blocking anti-integrin α3 (clone P1B5), anti-integrin α5 (clone P1D6), anti-integrin β1 (clone 6S6), or anti-integrin β3 (clone B3A) mouse mAb (all from Millipore). Controls were incubated with cell culture medium alone. Subsequently, tumor cell adhesion to immobilized collagen, as well as chemotaxis, was evaluated as described above.

### 4.11. Statistics

Curcumin uptake, adhesion, chemotaxis, and migration experiments were performed six times, and statistical significance was determined with the Wilcoxon–Mann-Whitney-U-test. Western bloting was done three times and statistics evaluated by t-test. Values are means ± SD. Differences were considered statistically significant at a p-value less than 0.05.

## 5. Conclusions

Since it is technically feasible to apply visible light during tumor resection, combining curcumin application with visible light could enhance the RCC treatment protocol, and compensate for the low bioavailability and rapid degradation of curcumin. The data presented here indicate a curcumin uptake within 1 h. This time window should be considered during surgery, e.g., by infusing curcumin 1 h prior to light application in future RCC in vivo models.

## Figures and Tables

**Figure 1 cancers-12-00302-f001:**
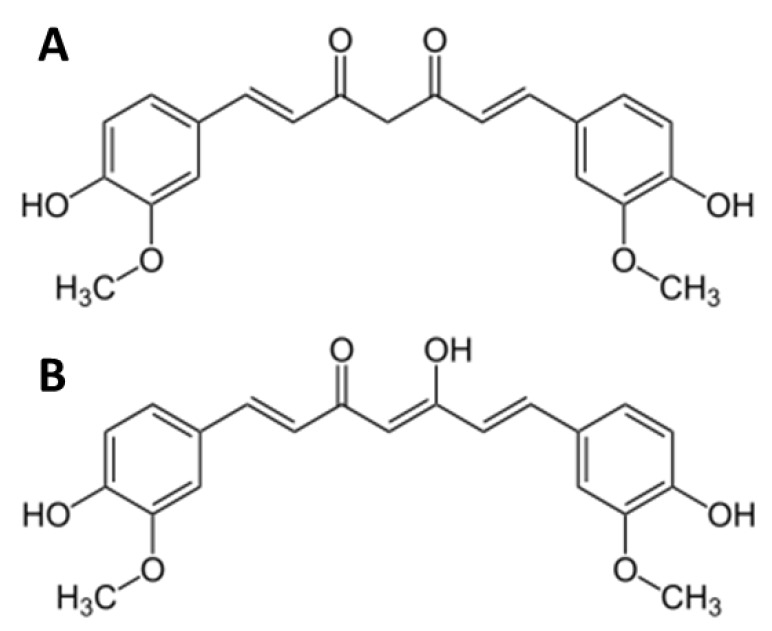
Chemical structure of curcumin (C_21_H_20_O_6_), (**A**) shows keto and (**B**) enol form [18].

**Figure 2 cancers-12-00302-f002:**
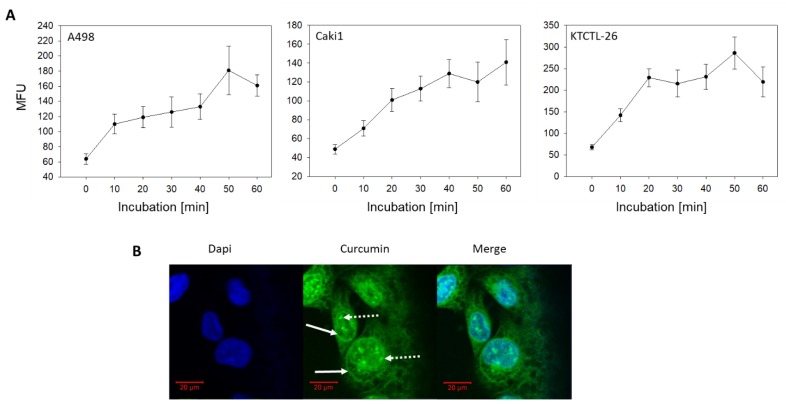
(**A**) Curcumin uptake in A498, Caki1, KTCTL-26 cells. Each value is the mean ± SD (standard deviation) of three independent experiments. (**B**) Intracellular distribution of curcumin (4 µg/mL) in A498 cells (representative for all three cell lines). Fluorescence shown by confocal laser-scanning microscopy after 60 min. Solid arrows: accumulation of curcumin along the nuclear membrane, dashed arrows: accumulation of curcumin within the nucleus. MFU = mean fluorescence units, DAPI = 4′,6-Diamidine-2′-phenylindole dihydrochloride.

**Figure 3 cancers-12-00302-f003:**
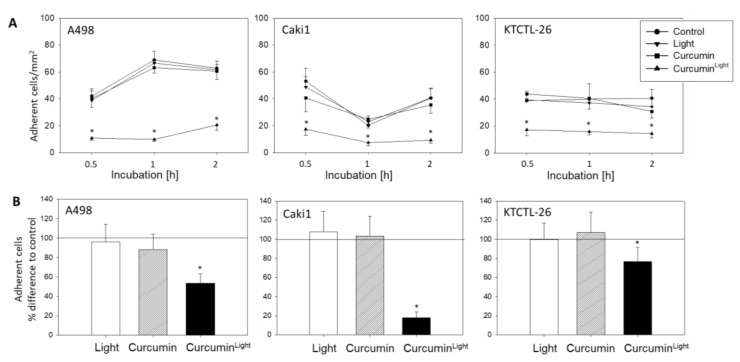
Influence of curcumin (0.2 µg/mL), light, or curcumin^Light^ on adhesion of RCC cells to HUVECs (**A**) and (**B**) collagen. Five separate fields of 0.25 mm^2^ were counted at 200× magnification (means ± SD, *n* = 6). Control was added cell medium and is indicated by the line at 100% in (**B**). * indicates significant difference to controls (*p* = 0.00512).

**Figure 4 cancers-12-00302-f004:**
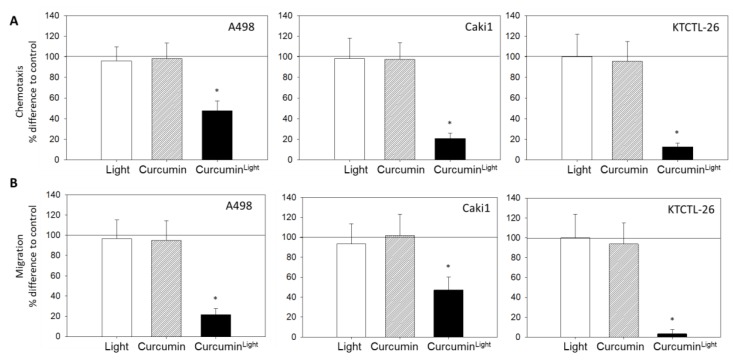
Influence of curcumin (0.2 µg/mL), light, or curcumin^Light^ on chemotaxis towards a serum gradient (**A**) and migration through a collagen matrix (**B**). Endpoints after 24 h. Untreated control cells were set to 100%, indicated by a line drawn at 100%. 5 separate fields of 0.25 mm^2^ were counted at 200× magnification (means ± SD, *n* = 6). * indicates significant difference to controls (*p* = 0.00512).

**Figure 5 cancers-12-00302-f005:**
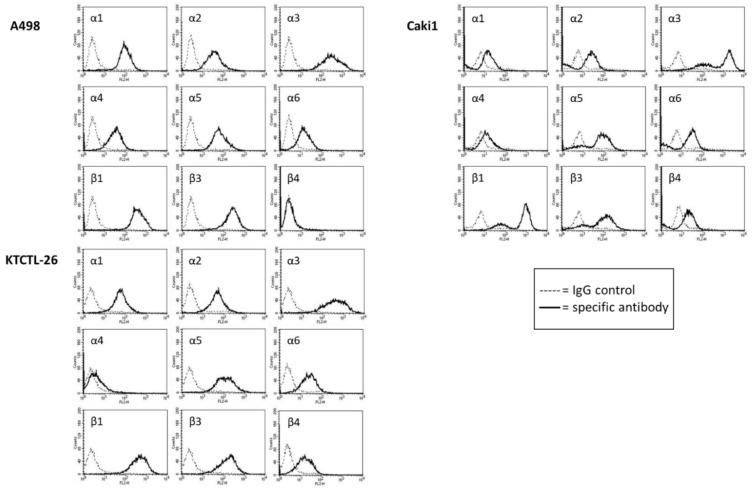
Surface expression of α and β integrins on A498, Caki1, and KTCTL-26 cells. Measured by Figure 1. PE, IgG2a-PE and IgG2b-PE (dashed line). The abscissa shows the relative logarithmic distribution of the relative fluorescence intensity of α1-α6 and β1, β3, and β4. The ordinate shows cell number. 10,000 cells were counted. Figure is representative for *n* = 6.

**Figure 6 cancers-12-00302-f006:**
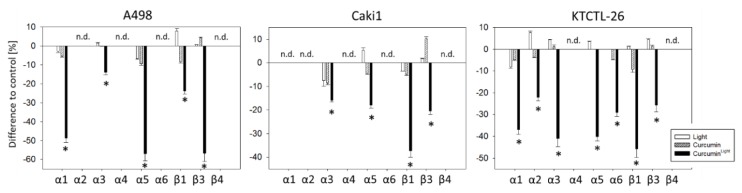
Influence of curcumin (0.2 µg/mL), light, or curcumin^Light^ on the integrin expression profile of A498, Caki1, and KTCTL-26 cells. The untreated control is set to 0% (line drawn at 0%). Values are means ± SD, *n* = 6. * indicates significant difference to controls (*p* = 0.00512). n.d. = not or hardly detectable.

**Figure 7 cancers-12-00302-f007:**
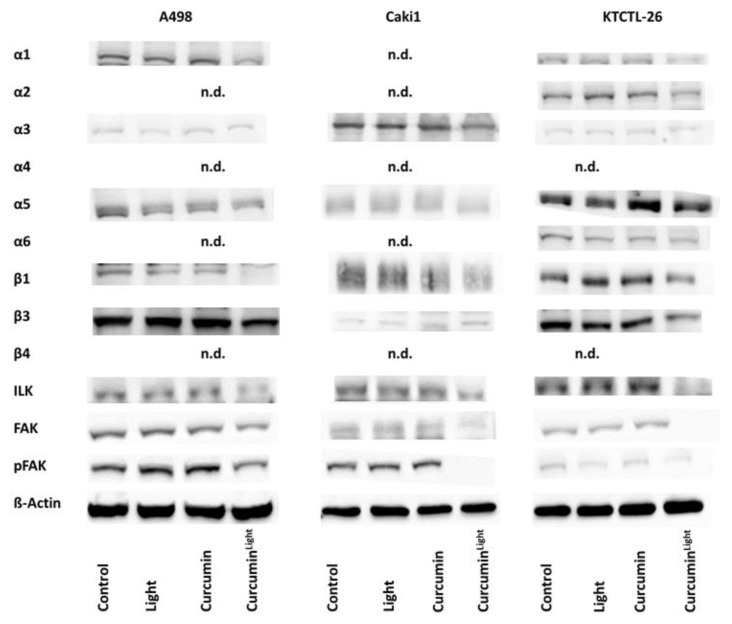
Western blot of α and β integrins, ILK and pFAK depending on the influence of curcumin (0.2 µg/mL), light, and curcumin^Light^ on A498, Caki1, and KTCTL-26 cells. Protein levels were measured 24 h after respective treatments. All bands are representative of *n* = 3. β-actin served as loading control and is representatively shown once. 50 µg were used per sample. n.d. = not detectable.

**Figure 8 cancers-12-00302-f008:**
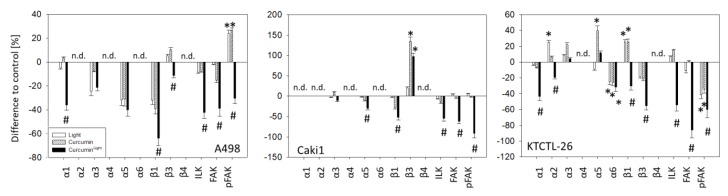
Pixel density of Western blot of α and β integrins, ILK and pFAK, depending on the influence of curcumin (0.2 µg/mL), light, or curcumin^Light^ on A498, Caki1, and KTCTL-26 cells. Values are means ± SD, *n* = 3. * indicates significant difference to the untreated control (line drawn at 0%) and # indicates significant difference to light alone or curcumin alone. n.d. = not detectable.

**Figure 9 cancers-12-00302-f009:**
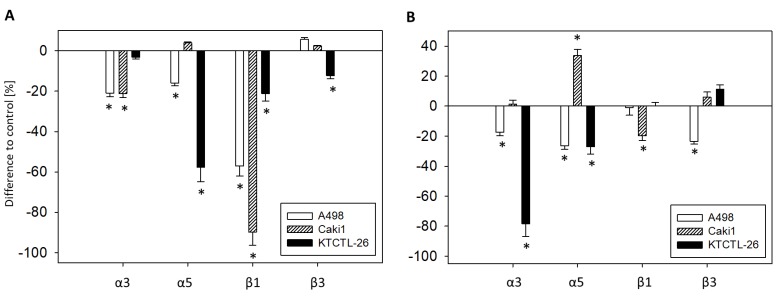
Adhesion to collagen (**A**) and chemotaxis (**B**) of A498, Caki1, and KTCTL-26 cells after blockade of integrins α3, α5, β1, or β3. The untreated control is set to 0% (line drawn at 0%). 5 separate fields of 0.25 mm^2^ were counted at 200× magnification (means ± SD, *n* = 6). * indicates significant difference to controls (*p* = 0.00512).

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
