# Peer review of "Low Dosed Curcumin Combined with Visible Light Exposure Inhibits Renal Cell Carcinoma Metastatic Behavior in Vitros"

_cancers, 2020, doi:10.3390/cancers12020302_

Round 1

Reviewer 1 Report

“Low dosed curcumin combined with visible light exposure inhibits renal cell carcinoma metastatic behavior in vitro”

Authors: Jochen Rutz , Sebastian Maxeiner , Saira Justin , Beatrice Bachmeier , August Bernd , Stefan Kippenberger , Nadja Zöller , Felix Chun , and Roman Blaheta

Summary

Tumor cell chemotaxis, migration and adhesion are key procedures involved in cancer metastasis and correlated with poor outcome. Advancements in the development of molecular therapeutic targeting of these pathological factors have made precision treatment possible in the management of advanced cancers nowadays. Curcumin has been shown to exert a protective role against tumor proliferation and migration through the regulation of multiple significant molecular targets, such as cell cycle proteins, receptors, and cell surface adhesion molecules. The combination of low-dose curcumin with visible light irradiation have been shown to inhibit growth and proliferation of renal cancer cells, but the role of this combined strategy in tumor metastasis is unknown. Thus, the presented report investigated the potential beneficial role of low-dose curcumin in conjunction with visible light on adhesion, migration, and chemotaxis in three kidney cancer cell lines in an in vitro model of metastatic behavior.

Comments

With rapid population aging and unhealthy lifestyles, the number of cancer patients is expected to continue increasing. Therefore, investigating novel strategies for targeting protective signal pathways that control migration and invasion of cancer cells are interesting.

However, the proposed study has several shortcomings.  

Line 52, reference 8 deals with issues of spinal disc, not issues in cancer. Introduction: The following needs to be addressed. a) interaction of curcumin in conjunction with visible light was already explored by their group (Jochen Rutz et al. 2019) and b) rationale of this combined strategy against tumor metastasis in the current study. Materials and Methods, The rationale for dose selection of a single concentration of 4 µg/ml and 0.54 µM of curcumin in distribution and other in vitro experiment should be indicated, as well as irradiation dose of visible light (Line 299-300 in the manuscript). Experiments with dose responses could be conducted to address respective and combined impacts. The concentration of curcumin used in distribution and other in vitro experiment is inconsistent. I would not call bar graph of control in Figures 3,4,6,8. Since curcumin with light showed integrin attenuation to different extents, the efficiency of respective integrin blockage should be included for comparison. The unexpected effect between α5 surface expression and Caki1 chemotaxis has been well interpreted in the text. The mechanism underlying the protective curcumin plus light against metastatic behavior will have to be verified using in vivo models.

Additional issues.

The abstract seems verbose and should be concise. Lines 44: Typing errors were found with ‘an”. Line 53-54 : descriptions of curcumin are the same as the publication, Int J Mol Sci. 2019 Mar; 20(6): 1464 (the fifth line of the second paragraph, introduction section).  Line 102 : Typing errors were found with “(SD)”.

Reviewer 2 Report

The authors bring important and novel information regarding the curcumin therapeutic benefits, describing molecular mechanisms influenced by curcumin treatment in renal cell carcinoma (in vitro study).

The following corrections I consider important to be made:

Introduction

-please offer a reference about the affirmation in line 42(... 20-30% developing metastases during therapy)

no refference for the affirmation in line 42-43 please see the study "Comparative effect of curcumin versus liposomal curcumin on systemic pro-inflammatory cytokine profile, MCP-1 and RANTES in experimental diabetes mellitus, International Journal of Nanomedicine, 2019, 14:8961-89-72" for curcumin liposomal formulation in order to incrase their bioavailability, since you made  this affirmation in line 63-66. sentence in line 72-75 is to long and difficult to read. Plese wrap it or reformulate it.

Results

there are no data regarding the mean and standard deviations of the parameteres they assessed, which is unacceptable. The authors presented their results only in figures which is insufficient. There are also no " P " values provided, even they are mentioning significant differences between groups. Plese provide all the results as mean,  standard deviations and P values. Otherwise I consider this paper undesirable for publication. 

Disscusion

Please describe more detailed the mechanism by which the curcumin effect is enhanced by visible light and some references regarding this properties.

Line 232-233: "the beta3 protein increase in Caki1 is interpreted such that curculin -light induces translocation from outside the cell into the cytoplasm" - this affirmation should be interpreted as a hypothesis

Line 250: " an inverse correlation between alpha5 expression and Caki1 chemotaxis is paradoxic...". How they have this conclusion since there is no statistical significant correlations described in this study?

Conclusions

They should explain better how they consider that exposure of tumor bed to light and the curcumin solution administration can be techical applied, since they found that the curcumin was incorporated into the cells in 40 and 50 min respectively (line 78-79).

I consider this paper a very important aquisition for curcumin therapeutic properties study and I would like to see it published.

Round 2

Reviewer 1 Report

All the issues raised have been addressed. 

Reviewer 2 Report

Cancer is a major challenge worldwide therefore any scientific achievments that can contribute to the improvements of treatment modalities is important. Very interesting study about curcumin efficiency in cancer metastatic molecular mechanisms. Taken together the results in this paper may open a way for a useful approach to combat cancer.

I would like to see this paper published due to its interest in cancer research field. Congratulation to the team for their contribution to the scientific knowledge about metastasis and curcumin properties and efficiency for diminishing of this process.